# Sparse Uncertainty Representation in Deep Learning with Inducing Weights

## Abstract

Bayesian neural networks and deep ensembles represent two modern paradigms of uncertainty quantification in deep learning. Yet these approaches struggle to scale mainly due to memory inefficiency issues, since they require parameter storage several times higher than their deterministic counterparts. To address this, we augment the weight matrix of each layer with a small number of inducing weights, thereby projecting the uncertainty quantification into such low dimensional spaces. We further extend Matheron's conditional Gaussian sampling rule to enable fast weight sampling, which enables our inference method to maintain reasonable run-time as compared with ensembles. Importantly, our approach achieves competitive performance to the state-of-the-art in prediction and uncertainty estimation tasks with fully connected neural networks and ResNets, while reducing the parameter size to $\leq 47.9\%$ of that of a single neural network.

## 1 Introduction

Deep learning models are becoming deeper and wider than ever before. From image recognition models such as ResNet-101 (He et al., 2016a) and DenseNet (Huang et al., 2017) to BERT (Xu et al., 2019) and GPT-3 (Brown et al., 2020) for language modelling, deep neural networks have found consistent success in fitting large-scale data. As these models are increasingly deployed in real-world applications, calibrated uncertainty estimates for their predictions become crucial, especially in safety-critical areas such as healthcare. In this regard, Bayesian neural networks (BNNs) (MacKay, 1995; Blundell et al., 2015; Gal & Ghahramani, 2016; Zhang et al., 2020) and deep ensembles (Lakshminarayanan et al., 2017) represent two popular paradigms for estimating uncertainty, which have shown promising results in applications such as (medical) image processing (Kendall & Gal, 2017; Tanno et al., 2017) and out-of-distribution detection (Ovadia et al., 2019).

Though progress has been made, one major obstacle to scaling up BNNs and deep ensembles is the computation cost in both time and space complexities. Especially for the latter, both approaches require the number of parameters to be several times higher than their deterministic counterparts. Recent efforts have been made to improve their memory efficiency (Louizos & Welling, 2017; Swiatkowski et al., 2020; Wen et al., 2020; Dusenberry et al., 2020). Still, these approaches require storage memory that is higher than storing a deterministic neural network.

Perhaps surprisingly, when taking the width of the network layers to the infinite limit, the resulting neural network becomes "parameter efficient". Indeed, an infinitely wide BNN becomes a Gaussian process (GP) that is known for good uncertainty estimates (Neal, 1995; Matthews et al., 2018; Lee et al., 2018). Effectively, the "parameters" of a GP are the datapoints, which have a considerably smaller memory footprint. To further reduce the computational burden, sparse posterior approximations with a small number of inducing points are widely used (Snelson & Ghahramani, 2006; Titsias, 2009), rendering sparse GPs more memory efficient than their neural network counterparts.

Can we bring the advantages of sparse approximations in GPs – which are infinitely-wide neural networks – to finite width deep learning models? We provide an affirmative answer regarding memory efficiency, by proposing an uncertainty quantification framework based on sparse uncertainty representations. We present our approach in BNN context, but the proposed approach is applicable to deep ensembles as well. In details, our contributions are as follows:

- We introduce underline{inducing weights} as auxiliary variables for uncertainty estimation in deep neural networks with efficient (approximate) posterior sampling. Specifically:

  - We introduce inducing weights — lower dimensional counterparts to the actual weight matrices — for variational inference in BNNs, as well as a memory efficient parameterisation and an extension to ensemble methods (Section 3.1).
  - We extend Matheron's rule to facilitate efficient posterior sampling (Section 3.2).
  - We show the connection to sparse (deep) GPs, in that inducing weights can be viewed as underline{projected noisy inducing outputs} in pre-activation output space (Section 3.3).
  - We provide an in-depth computation complexity analysis (Section 3.4), showing the significant advantage in terms of parameter efficiency.

- We apply the proposed approach to both BNNs and deep ensembles. Experiments in classification, model robustness and out-of-distribution detection tasks show that our inducing weight approaches achieve competitive performance to their counterparts in the original weight space on modern deep architectures for image classification, while reducing the parameter count to less than half of that of a single neural network.

## 2    VARIATIONAL INFERENCE WITH INDUCING VARIABLES

This section lays out the basics on variational inference and inducing variables for posterior approximations, which serve as foundation and inspiration for this work. Given observations $\mathcal{D} = \{\mathbf{X}, \mathbf{Y}\}$ with $\mathbf{X} = [\boldsymbol{x}_1, ..., \boldsymbol{x}_N]$, $\mathbf{Y} = [\boldsymbol{y}_1, ..., \boldsymbol{y}_N]$, we would like to fit a neural network $p(\boldsymbol{y}|\boldsymbol{x}, W_{1:L})$ with weights $W_{1:L}$ to the data. BNNs posit a prior distribution $p(W_{1:L})$ over the weights, and construct an approximate posterior $q(W_{1:L})$ to the intractable exact posterior $p(W_{1:L}|\mathcal{D}) \propto p(\mathcal{D}|W_{1:L})p(W_{1:L})$, where $p(\mathcal{D}|W_{1:L}) = p(\mathbf{Y}|\mathbf{X}, W_{1:L}) = \prod_{n=1}^{N} p(\boldsymbol{y}_n|\boldsymbol{x}_n, W_{1:L})$.

**Variational inference**    Variational inference (Jordan et al., 1999; Zhang et al., 2018a) constructs an approximation $q(\theta)$ to the posterior $p(\theta|\mathcal{D}) \propto p(\theta)p(\mathcal{D}|\theta)$ by maximising a variational lower-bound:

$$\log p(\mathcal{D}) \geq \mathcal{L}(q(\theta)) := \mathbb{E}_{q(\theta)}\left[\log p(\mathcal{D}|\theta)\right] - \mathbb{KL}\left[q(\theta)||p(\theta)\right]. \tag{1}$$

For BNNs, $\theta = \{W_{1:L}\}$, and a simple choice of $q$ is a fully factorised Gaussian (FFG): $q(W_{1:L}) = \prod_{l=1}^{L} \prod_{i=1}^{d_{out}^l} \prod_{j=1}^{d_{in}^l} \mathcal{N}(m_l^{(i,j)}, v_l^{(i,j)})$, with $m_l^{(i,j)}, v_l^{(i,j)}$ the mean and variance of $W_l^{(i,j)}$ and $d_{in}^l, d_{out}^l$ the respective number of inputs and outputs to layer $l$. The variational parameters are then $\phi = \{m_l^{(i,j)}, v_l^{(i,j)}\}_{l=1}^{L}$. Gradients of (1) w.r.t. $\phi$ can be estimated with mini-batches of data (Hoffman et al., 2013) and with Monte Carlo sampling from the $q$ distribution (Titsias & Lázaro-Gredilla, 2014; Kingma & Welling, 2014). By setting $q$ to an FFG, a variational BNN can be trained with similar computational requirements as a deterministic network (Blundell et al., 2015).

**Improved posterior approximation with inducing variables**    Auxiliary variable approaches (Agakov & Barber, 2004; Salimans et al., 2015; Ranganath et al., 2016) construct the $q(\theta)$ distribution with an auxiliary variable $a$: $q(\theta) = \int q(\theta|a)q(a)da$, with the hope that a potentially richer mixture distribution $q(\theta)$ can achieve better approximations. As then $q(\theta)$ becomes intractable, an auxiliary variational lower-bound is used to optimise $q(\theta, a)$:

$$\log p(\mathcal{D}) \geq \mathcal{L}(q(\theta, a)) = \mathbb{E}_{q(\theta, a)}[\log p(\mathcal{D}|\theta)] + \mathbb{E}_{q(\theta, a)}\left[\log \frac{p(\theta)r(a|\theta)}{q(\theta|a)q(a)}\right]. \tag{2}$$

Here $r(a|\theta)$ is an auxiliary distribution that needs to be specified, where existing approaches often use a "reverse model" for $r(a|\theta)$. Instead, we define $r(\theta|a)$ in a generative manner: $r(a|\theta)$ is the "posterior" of the following "generative model", whose "evidence" is exactly the prior of $\theta$:

$$r(a|\theta) = \tilde{p}(a|\theta) \propto \tilde{p}(a)\tilde{p}(\theta|a), \text{ such that } \tilde{p}(\theta) := \int \tilde{p}(a)\tilde{p}(\theta|a)da = p(\theta). \tag{3}$$

Plugging in (3) to (2) immediately leads to:

$$\mathcal{L}(q(\theta, a)) = \mathbb{E}_{q(\theta)}[\log p(\mathcal{D}|\theta)] - \mathbb{E}_{q(a)}\left[\mathbb{KL}[q(\theta|a)||\tilde{p}(\theta|a)]\right] - \mathbb{KL}[q(a)||\tilde{p}(a)]. \tag{4}$$

This approach returns an efficient approximate inference algorithm, translating the complexity of inference in $\theta$ to $a$, if $\dim(a) < \dim(\theta)$ and $q(\theta, a) = q(\theta|a)q(a)$ has the following properties:

1. A "pseudo prior" $\tilde{p}(a)\tilde{p}(\theta|a)$ is defined such that $\int \tilde{p}(a)\tilde{p}(\theta|a)da = p(\theta)$;

2. The conditional distributions $q(\theta|a)$ and $\tilde{p}(\theta|a)$ are in the same parametric family, so that they can share parameters;

3. Both sampling $\theta \sim q(\theta)$ and computing $\mathbb{KL}[q(\theta|a)||\tilde{p}(\theta|a)]$ can be done efficiently;

4. The designs of $q(a)$ and $\tilde{p}(a)$ can potentially provide extra advantages (in time and space complexities and/or optimisation easiness).

We call $a$ the inducing variable of $\theta$, which is inspired by varationally sparse GP (SVGP) with inducing points (Snelson & Ghahramani, 2006; Titsias, 2009). Indeed SVGP is a special case: $\theta = \mathbf{f}$, $a = \mathbf{u}$, the GP prior is $p(\mathbf{f}|\mathbf{X}) = \mathcal{GP}(\mathbf{0}, \mathbf{K}_{\mathbf{XX}})$, $p(\mathbf{u}) = \mathcal{GP}(\mathbf{0}, \mathbf{K}_{\mathbf{ZZ}})$, $\tilde{p}(\mathbf{f}, \mathbf{u}) = p(\mathbf{u})p(\mathbf{f}|\mathbf{X}, \mathbf{u})$, $q(\mathbf{f}|\mathbf{u}) = p(\mathbf{f}|\mathbf{X}, \mathbf{u})$, $q(\mathbf{f}, \mathbf{u}) = p(\mathbf{f}|\mathbf{X}, \mathbf{u})q(\mathbf{u})$, and $\mathbf{Z}$ are the optimisable inducing inputs. The variational lower-bound is $\mathcal{L}(q(\mathbf{f}, \mathbf{u})) = \mathbb{E}_{q(\mathbf{f})}[\log p(\mathbf{Y}|\mathbf{f})] - \mathbb{KL}[q(\mathbf{u})||p(\mathbf{u})]$, and the variational parameters are $\phi = \{\mathbf{Z}, \text{distribution parameters of } q(\mathbf{u})\}$. SVGP satisfies the marginalisation constraint (3) by definition, and it has $\mathbb{KL}[q(\mathbf{f}|\mathbf{u})||\tilde{p}(\mathbf{f}|\mathbf{u})] = 0$. Also by using small $M = \dim(\mathbf{u})$ and exploiting the $q$ distribution design, SVGP reduces run-time from $\mathcal{O}(N^3)$ to $\mathcal{O}(NM^2)$ where $N$ is the number of inputs in $\mathbf{X}$, meanwhile it also makes storing a full Gaussian $q(\mathbf{u})$ affordable. Lastly, $\mathbf{u}$ can be whitened, leading to the "pseudo prior" $\tilde{p}(\mathbf{f}, \mathbf{v}) = p(\mathbf{f}|\mathbf{X}, \mathbf{u} = \mathbf{K}_{\mathbf{ZZ}}^{1/2}\mathbf{v})\tilde{p}(\mathbf{v}), \tilde{p}(\mathbf{v}) = \mathcal{N}(\mathbf{v}; \mathbf{0}, \mathbf{I})$ which could bring potential benefits in optimisation.

In the rest of the paper we assume the "pseudo prior" $\tilde{p}(\theta, a)$ satisfies the marginalisation constraint (3), allowing us to write $p(\theta, a) := \tilde{p}(\theta, a)$. It might seem unclear how to design $\tilde{p}(\theta, a)$ for an arbitrary probabilistic model, however, for a Gaussian prior on $\theta$ the rules for computing conditional Gaussian distributions can be used to construct $\tilde{p}$. In section 3 we exploit these rules to develop an efficient approximate inference method for Bayesian neural networks with inducing weights.

# 3 SPARSE UNCERTAINTY REPRESENTATION WITH INDUCING WEIGHTS

## 3.1 INDUCING WEIGHTS FOR NEURAL NETWORK PARAMETERS

Following the design principles of inducing variables, we introduce to each network layer $l$ a smaller inducing weight matrix $U_l$, and construct joint approximate posterior distributions for inference. In the rest of the paper we assume a factorised prior across layers $p(W_{1:L}) = \prod_l p(W_l)$, and for notation ease we drop the $l$ indices when the context is clear.

**Augmenting network layers with inducing weights** Suppose the weight $W \in \mathbb{R}^{d_{out} \times d_{in}}$ has a Gaussian prior $p(W) = p(\text{vec}(W)) = \mathcal{N}(0, \sigma^2 I)$ where $\text{vec}(W)$ concatenates the columns of the weight matrix into a vector. A first attempt to augment $p(\text{vec}(W))$ with an inducing weight variable $U \in \mathbb{R}^{M_{out} \times M_{in}}$ may be to construct a multivariate Gaussian $p(\text{vec}(W), \text{vec}(U))$, such that $\int p(\text{vec}(W), \text{vec}(U))dU = \mathcal{N}(0, \sigma^2 I)$. This means for the joint covariance matrix of $(\text{vec}(W), \text{vec}(U))$, it requires the block corresponding to the covariance of $\text{vec}(W)$ to match the prior covariance $\sigma^2 I$. We are then free to parameterise the rest of the entries in the joint covariance matrix, as long as this full matrix remains positive definite. Now the conditional distribution $p(W|U)$ is a function of these parameters, and the conditional sampling from $p(W|U)$ is further discussed in Appendix A.1. Unfortunately, as $\dim(\text{vec}(W))$ is typically large (e.g. of the order of $10^7$), using a full covariance Gaussian for $p(\text{vec}(W), \text{vec}(U))$ becomes computationally intractable.

This issue can be addressed using matrix normal distributions (Gupta & Nagar, 2018). Notice that the prior $p(\text{vec}(W)) = \mathcal{N}(\mathbf{0}, \sigma^2 I)$ has an equivalent matrix normal distribution form as $p(W) = \mathcal{MN}(0, \sigma_r^2 I, \sigma_c^2 I)$, with $\sigma_r, \sigma_c > 0$ the row and column standard deviations satisfying $\sigma = \sigma_r \sigma_c$. Now we introduce the inducing variables in matrix space, in addition to $U$ we pad in two auxiliary variables $U_r \in \mathbb{R}^{M_{out} \times d_{in}}, U_c \in \mathbb{R}^{d_{out} \times M_{in}}$, so that the full augmented prior is:

$$\begin{pmatrix} W & U_c \\ U_r & U \end{pmatrix} \sim p(W, U_c, U_r, U) := \mathcal{MN}(0, \Sigma_r, \Sigma_c), \tag{5}$$

$$\text{with} \quad L_r = \begin{pmatrix} \sigma_r I & 0 \\ Z_r & D_r \end{pmatrix} \quad \text{s.t.} \quad \Sigma_r = L_r L_r^\top = \begin{pmatrix} \sigma_r^2 I & \sigma_r Z_r^\top \\ \sigma_r Z_r & Z_r Z_r^\top + D_r^2 \end{pmatrix},$$

$$\text{and} \quad L_c = \begin{pmatrix} \sigma_c I & 0 \\ Z_c & D_c \end{pmatrix} \quad \text{s.t.} \quad \Sigma_c = L_c L_c^\top = \begin{pmatrix} \sigma_c^2 I & \sigma_c Z_c^\top \\ \sigma_c Z_c & Z_c Z_c^\top + D_c^2 \end{pmatrix}.$$

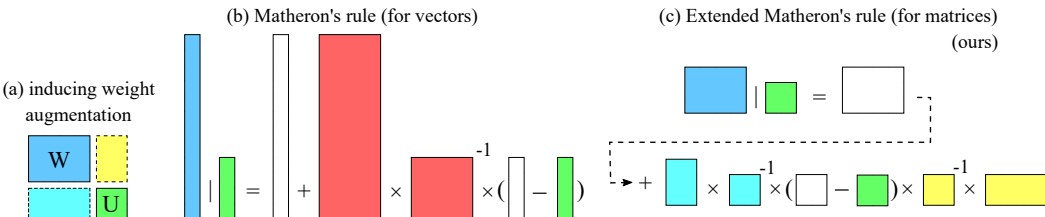

Figure 1: Visualisation of (a) the inducing weight augmentation, and compare (b) the original Matheron's rule to (c) our extended version. The white blocks represent random noises from the joint.

See Figure 1(a) for a visualisation. Matrix normal distributions have similar marginalization and conditioning properties as multivariate Gaussian distributions. Therefore the marginalisation constraint (3) is satisfied for any $Z_c, Z_r, D_c$ and $D_r$. The marginal distribution of the inducing weight is $p(U) = \mathcal{MN}(0, \Psi_r, \Psi_c)$ with $\Psi_r = Z_r Z_r^\top + D_r^2$ and $\Psi_c = Z_c Z_c^\top + D_c^2$. In the experiments we use whitened inducing weights which transforms $U$ so that $p(U) = \mathcal{MN}(0, I, I)$ (Appendix E), but for clarity, we continue using the formulas presented above in the main text.

The matrix normal parameterisation introduces two additional variables $U_r, U_c$ without providing additional expressiveness. Hence it is desirable to integrate them out, leading to a joint multivariate normal with Khatri-Rao product structure for the covariance:

$$p(\text{vec}(W), \text{vec}(U)) = \mathcal{N}\left(0, \begin{pmatrix} \sigma_c^2 I \otimes \sigma_r^2 I & \sigma_c Z_c^\top \otimes \sigma_r Z_r^\top \\ \sigma_c Z_c \otimes \sigma_r Z_r & \Psi_c \otimes \Psi_r \end{pmatrix}\right). \quad (6)$$

As the dominating memory complexity here is $\mathcal{O}(d_{out} M_{out} + d_{in} M_{in})$ which comes from storing $Z_r$ and $Z_c$, we see that the matrix normal parameterisation of the augmented prior is memory efficient.

**Posterior approximation in the joint space** We construct a factorised posterior approximation across the layers: $q(W_{1:L}, U_{1:L}) = \prod_l q(W_l|U_l) q(U_l)$.

The simplest option for $q(W|U)$ is $q(W|U) = p(\text{vec}(W)|\text{vec}(U)) = \mathcal{N}(\mu_{W|U}, \Sigma_{W|U})$, similar to sparse GPs. A slightly more flexible variant rescales the covariance matrix while keeping the mean tied, i.e. $q(W|U) = q(\text{vec}(W)|\text{vec}(U)) = \mathcal{N}(\mu_{W|U}, \lambda^2 \Sigma_{W|U})$, which still allows for the KL term to be calculated efficiently (see Appendix B):

$$R(\lambda) := \mathbb{KL}\left[q(W|U)||p(W|U)\right] = d_{in} d_{out}(0.5\lambda^2 - \log \lambda - 0.5), \quad W \in \mathbb{R}^{d_{out} \times d_{in}}. \quad (7)$$

Plugging $\theta = \{W_{1:L}\}, a = \{U_{1:L}\}$ into (4) results in the following variational lower-bound

$$\mathcal{L}(q(W_{1:L}, U_{1:L})) = \mathbb{E}_{q(W_{1:L})}[\log p(\mathcal{D}|W_{1:L})] - \sum_{l=1}^{L}(R(\lambda_l) + \mathbb{KL}[q(U_l)||p(U_l)]), \quad (8)$$

with $\lambda_l$ the associated scaling parameter for $q(W_l|U_l)$. Therefore the variational parameters are now $\phi = \{Z_c, Z_r, D_c, D_r, \lambda, \text{dist. params. of } q(U)\}$ for each network layer.

**Two choices of** $q(U)$ A simple choice is FFG $q(\text{vec}(U)) = \mathcal{N}(\boldsymbol{m}_u, \text{diag}(\boldsymbol{v}_u))$, which performs mean-field inference in $U$ space (c.f. Blundell et al., 2015), and in this case $\mathbb{KL}[q(U)||p(U)]$ has a closed-form solution. Another choice is a "mixture of delta measures" $q(U) = \frac{1}{K} \sum_{k=1}^{K} \delta(U = U^{(k)})$. In other words, we keep $K$ distinct sets of parameters $\{U_{1:L}^{(k)}\}_{k=1}^{K}$ in inducing space that are projected back into the original parameter space via the shared conditional distributions $q(W_l|U_l)$ to obtain the weights. This approach can be viewed as constructing "deep ensembles" in $U$ space, and we follow ensemble methods (e.g. Lakshminarayanan et al., 2017) to drop $\mathbb{KL}[q(U)||p(U)]$ in (8).

Often the inducing weight $U$ is chosen to have significantly lower dimensions compared to $W$. Combining with the fact that $q(W|U)$ and $p(W|U)$ only differ in the covariance scaling constant, we see that $U$ can be regarded as a sparse representation of uncertainty for the network layer, as the major updates in (approximate) posterior belief is quantified by $q(U)$.

### 3.2 Efficient sampling with extended Matheron's rule

Computing the variational lower-bound (8) requires samples from $q(W)$, which asks for an efficient sampling procedure for the conditional $q(W|U)$. Unfortunately, $q(W|U)$ derived from (6) with

Figure 2: Showing the $U$ variables in pre-activation spaces. To simplify we set $\sigma_c = 1$ w.l.o.g.

covariance rescaling is not a matrix normal, so direct sampling remains prohibitively expensive. To address this challenge, we extend Matheron's rule (Journel & Huijbregts, 1978; Hoffman & Ribak, 1991; Doucet, 2010) to efficiently sample from $q(W|U)$. The idea is that one can sample from a conditional Gaussian by transforming a sample from the joint distribution. In detail, we derive in Appendix C the extended Matheron's rule to sample $W \sim q(W|U)$:

$$W = \lambda \bar{W} + \sigma Z_r^\top \Psi_r^{-1}(U - \lambda \bar{U})\Psi_c^{-1}Z_c, \quad \bar{W}, \bar{U} \sim p(\bar{W}, \bar{U}_c, \bar{U}_r, \bar{U}) = \mathcal{MN}(0, \Sigma_r, \Sigma_c). \quad (9)$$

Here $\bar{W}, \bar{U} \sim p(\bar{W}, \bar{U}_c, \bar{U}_r, \bar{U})$ means we sample $\bar{W}, \bar{U}_c, \bar{U}_r, \bar{U}$ from the joint and drop $\bar{U}_c, \bar{U}_r$. In fact $\bar{U}_c, \bar{U}_r$ are never computed: as shown in Appendix C, the samples $\bar{W}, \bar{U}$ can be obtained by:

$$\bar{W} = \sigma E_1, \; \bar{U} = Z_r E_1 Z_c^\top + \hat{L}_r \tilde{E}_2 D_c + D_r \tilde{E}_3 \hat{L}_c^\top + D_r E_4 D_c, \; E_1 \sim \mathcal{MN}(0, I_{d_{out}}, I_{d_{in}}),$$
$$\tilde{E}_2, \tilde{E}_3, E_4 \sim \mathcal{MN}(0, I_{M_{out}}, I_{M_{in}}), \; \hat{L}_r = \text{Cholesky}(Z_r Z_r^\top), \; \hat{L}_c = \text{Cholesky}(Z_c Z_c^\top). \quad (10)$$

Therefore the major extra cost to pay is $\mathcal{O}(2M_{out}^3 + 2M_{in}^3 + d_{out}M_{out}M_{in} + M_{in}d_{out}d_{in})$ required by inverting $\Psi_r, \Psi_c$, computing $\hat{L}_r, \hat{L}_c$, and the matrix multiplications. The extended Matheron's rule is visualised in Figure 1 with a comparison to the original Matheron's rule for sampling from $q(\text{vec}(W)| \text{vec}(U))$. This clearly shows that our recipe avoids computing big matrix inverses and multiplications, resulting in a significant speed-up for conditional sampling.

### 3.3 Understanding inducing weights: a function-space perspective

We present the proposed approach again but from a function-space inference perspective. Assume a network layer computes the following transformation of the input $\mathbf{X} = [\boldsymbol{x}_1, ..., \boldsymbol{x}_N], \boldsymbol{x}_i \in \mathbb{R}^{d_{in} \times 1}$:

$$\mathbf{F} = W\mathbf{X}, \; \mathbf{H} = g(\mathbf{F}), \quad W \in \mathbb{R}^{d_{out} \times d_{in}}, \mathbf{X} \in \mathbb{R}^{d_{in} \times N}, g(\cdot) \text{ is the non-linearity.} \quad (11)$$

As $W$ has a Gaussian prior $p(\text{vec}(W)) = \mathcal{N}(0, \sigma^2 I)$, each of the rows in $\mathbf{F} = [\mathbf{f}_1, ..., \mathbf{f}_{d_{out}}]^\top, \mathbf{f}_i \in \mathbb{R}^{N \times 1}$ has a Gaussian process form with linear kernel: $\mathbf{f}_i | \mathbf{X} \sim \mathcal{GP}(\mathbf{0}, \mathbf{K_{XX}}), \mathbf{K_{XX}}(m, n) = \sigma^2 \boldsymbol{x}_m^\top \boldsymbol{x}_n$. Inference on $\mathbf{F}$ directly has $\mathcal{O}(N^3 + d_{out}N^2)$ cost, so a sparse approximation is needed. Slightly different from the usual approach, we introduce auxiliary variables $U_c = [\mathbf{u}_1^c, ..., \mathbf{u}_{d_{out}}^c]^\top \in \mathbb{R}^{d_{out} \times M_{in}}$ as follows, using shared "inducing inputs" $Z_c^\top \in \mathbb{R}^{d_{in} \times M_{in}}$:

$$p(\mathbf{f}_i, \hat{\mathbf{u}}_i | \mathbf{X}) = \mathcal{GP}\left(\mathbf{0}, \mathbf{K}_{[\mathbf{X}, Z_c^\top], [\mathbf{X}, Z_c^\top]}\right), \quad p(\mathbf{u}_i^c | \hat{\mathbf{u}}_i) = \mathcal{N}\left(\hat{\mathbf{u}}_i / \sigma_c, \sigma_r^2 D_c^2\right), \quad (12)$$

By marginalising out the "noiseless inducing outputs" $\{\hat{\mathbf{u}}_i\}$ (derivations in Appendix D), we can compute the marginal distributions as $p(U_c) := p(\{\mathbf{u}_i^c\}) = \mathcal{MN}(\mathbf{0}, \sigma_r^2 I, \Psi_c)$ and

$$p(\mathbf{F}|\mathbf{X}, U_c) = \mathcal{MN}\left(U_c \Psi_c^{-1} \sigma_c Z_c \mathbf{X}, \sigma_r^2 I, \mathbf{X}^\top \sigma_c^2(I - Z_c^\top \Psi_c^{-1} Z_c)\mathbf{X}\right). \quad (13)$$

Note that $p(W|U_c) = \mathcal{MN}\left(U_c \Psi_c^{-1} \sigma_c Z_c, \sigma_r^2 I, \sigma_c^2(I - Z_c^\top \Psi_c^{-1} Z_c)\right)$ (see Appendix A.2). Since $W \sim \mathcal{MN}(\mathbf{M}, \Sigma_1, \Sigma_2)$ leads to $W\mathbf{X} \stackrel{d}{\sim} \mathcal{MN}(\mathbf{MX}, \Sigma_1, \mathbf{X}^\top \Sigma_2 \mathbf{X})$, this means $p(\mathbf{F}|\mathbf{X}, U_c)$ is the push-forward distribution of $p(W|U_c)$ for the operation $\mathbf{F} = W\mathbf{X}$:

$$\mathbf{F} \sim p(\mathbf{F}|\mathbf{X}, U_c) \quad \Leftrightarrow \quad W \sim p(W|U_c), \mathbf{F} = W\mathbf{X}.$$

As $\{\mathbf{u}_i^c\}$ are the "noisy" versions of $\{\hat{\mathbf{u}}_i\}$ in $\mathbf{f}$ space, $U_c$ can thus be viewed as "scaled noisy inducing outputs" in function space (see the red bars in the 2nd column of Figure 2).

So far the inducing variables $U_c$ is used as a compact representation of the correlation between columns of $\mathbf{F}$ only. In other words the output dimension $d_{out}$ for each $\mathbf{f}_i$ (and $\mathbf{u}_i^c$) remains large (e.g. $> 1000$ in a fully connected layer). Therefore dimensionality reduction can be applied to the

Table 1: Computational complexity for a single layer. We assume $W \in \mathbb{R}^{d_{out} \times d_{in}}$, $U \in \mathbb{R}^{M_{out} \times M_{in}}$, and $K$ forward passes are made for each of the $N$ inputs. (*It uses a parallel computing friendly vectorisation technique (Wen et al., 2020) for further speed-up.)

| Method | Time complexity | Memory complexity |
|---|---|---|
| Deterministic-$W$ | $\mathcal{O}(N d_{in} d_{out})$ | $\mathcal{O}(d_{in} d_{out})$ |
| FFG-$W$ | $\mathcal{O}(N K d_{in} d_{out})$ | $\mathcal{O}(2 d_{in} d_{out})$ |
| Ensemble-$W$ | $\mathcal{O}(N K d_{in} d_{out})$ | $\mathcal{O}(K d_{in} d_{out})$ |
| Matrix-normal-$W$ | $\mathcal{O}(N K d_{in} d_{out})$ | $\mathcal{O}(d_{in} d_{out} + d_{in} + d_{out})$ |
| $k$-tied FFG-$W$ | $\mathcal{O}(N K d_{in} d_{out})$ | $\mathcal{O}(d_{in} d_{out} + k(d_{in} + d_{out}))$ |
| rank-1 BNN | $\mathcal{O}(N K d_{in} d_{out})^*$ | $\mathcal{O}(d_{in} d_{out} + 2(d_{in} + d_{out}))$ |
| FFG-$U$ | $\mathcal{O}(N K d_{in} d_{out} + 2M_{in}^3 + 2M_{out}^3$ $+ K(d_{out} M_{out} M_{in} + M_{in} d_{out} d_{in}))$ | $\mathcal{O}(d_{in} M_{in} + d_{out} M_{out} + 2 M_{in} M_{out})$ |
| Ensemble-$U$ | same as above | $\mathcal{O}(d_{in} M_{in} + d_{out} M_{out} + K M_{in} M_{out})$ |

column vectors of $U_c$ and $\mathbf{F}$. In Appendix D we present a generative approach to do so by extending probabilistic PCA (Tipping & Bishop, 1999) to matrix normals: $p(U_c) = \int p(U_c|U)p(U)dU$, where the projection's parameters are $\{Z_r, D_r\}$, and $p(U_c, U)$ matches the marginals of (5). This means $U$ can be viewed as the "noisy projected inducing output" of the GP whose corresponding "inducing inputs" are $Z_c^\top$ (see the red bar in the 1st column of Figure 2). Similarly the column vectors in $U_r\mathbf{X}$ can be viewed as the noisy projections of the column vectors in $\mathbf{F}$.

In Appendix D we further show that the push-forward distribution $q(\mathbf{F}|\mathbf{X}, U)$ of $q(W|U)$ only differs from $p(\mathbf{F}|\mathbf{X}, U)$ in the covariance matrices up to the same scaling constant $\lambda$. Therefore the resulting function-space variational objective is almost identical to (8), except for scaling coefficients that are added to the $R(\lambda_l)$ terms to account for the change in dimensionality from $\text{vec}(W)$ to $\text{vec}(\mathbf{F})$. This result nicely connects posterior inference in weight- and function-space.

## 3.4 COMPUTATIONAL COMPLEXITIES

In Table 1 we report the computational complexity figures for two types of inducing weight approaches: FFG $q(U)$ (FFG-$U$) and Delta mixture $q(U)$ (Ensemble-$U$). Baseline approaches include: Deterministic-$W$, variational inference with FFG $q(W)$ (FFG-$W$, Blundell et al., 2015), deep ensemble in $W$ (Ensemble-$W$, Lakshminarayanan et al., 2017), as well as parameter efficient approaches such as matrix-normal $q(W)$ (Matrix-normal-$W$, Louizos & Welling (2017)), variational inference with $k$-tied FFG $q(W)$ ($k$-tied FFG-$W$, Swiatkowski et al. (2020)), and rank-1 BNN (Dusenberry et al., 2020). The gain in memory is significant for the inducing weight approaches, in fact with $M_{in} < d_{in}$ and $M_{out} < d_{out}$ the parameter storage requirement is smaller than a single deterministic neural network. The major overhead in run-time comes from the extended Matheron's rule for sampling $q(W|U)$. Some of the computations there are performed only once, and in our experiments we show that by using a relatively low-dimensional $U$, the overhead is acceptable.

## 4 EXPERIMENTS

We evaluate the inducing weight approaches on regression, classification and related uncertainty estimation tasks. The goal is to demonstrate competitive performance to popular $W$-space uncertainty estimation methods while using significantly fewer parameters. The evaluation baselines are: (1) variational inference with FFG $q(W)$ (FFG-$W$, Blundell et al., 2015) v.s. FFG $q(U)$ (FFG-$U$, ours); (2) ensemble methods in $W$ space (Ensemble-$W$ Lakshminarayanan et al., 2017) v.s. ensemble in $U$ space (Ensemble-$U$, ours). Another baseline is training a deterministic neural network with maximum likelihood. Details and additional results can be found in Appendix F and G.

## 4.1 SYNTHETIC 1-D REGRESSION

We follow Foong et al. (2019) to construct a synthetic regression task, by sampling two clusters of inputs $x_1 \sim \mathcal{U}[-1, -0.7]$, $x_2 \sim \mathcal{U}[0.5, 1]$, and targets $y \sim \mathcal{N}(\cos(4x + 0.8), 0.01)$. As ground truth we show the exact posterior results using the NUTS sampler (Hoffman & Gelman, 2014). The results are visualised in Figure 3 with the noiseless function in black, predictive mean in blue, and

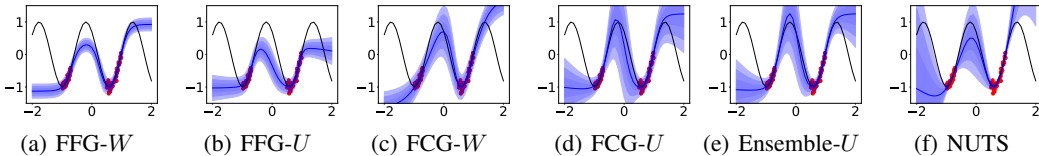

(a) FFG-$W$ (b) FFG-$U$ (c) FCG-$W$ (d) FCG-$U$ (e) Ensemble-$U$ (f) NUTS

Figure 3: Toy regression results, with observations in red dots and the ground truth function in black.

Table 2: CIFAR in-distribution metrics (in %).

| Method | CIFAR10 | | CIFAR100 | |
|---|---|---|---|---|
| | Acc. ↑ | ECE ↓ | Acc. ↑ | ECE ↓ |
| Deterministic-$W$ | 93.02 | 5.23 | 72.68 | 19.41 |
| Ensemble-$W$ | 94.94 | 1.25 | 76.61 | 6.25 |
| FFG-$W$ | 93.22 | 0.55 | 73.44 | 5.49 |
| FFG-$U$ | 91.52 | 1.31 | 75.69 | 5.20 |
| Ensemble-$U$ | 92.20 | 0.80 | 76.10 | 2.49 |

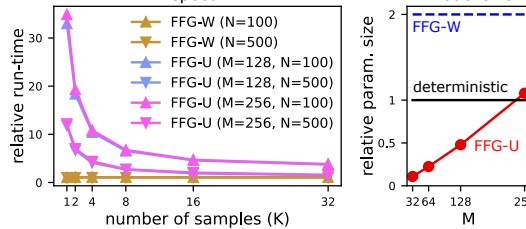

Figure 4: Resnet-18 run-times and model sizes.

up to three standard deviations as shaded area. Similar to prior results in the literature, FFG-$W$ fails to represent the increased uncertainty away from the data and in between clusters. While underestimating predictive uncertainty overall, FFG-$U$ show a small increase in predictive uncertainty away from the data. In contrast, a per-layer full covariance Gaussian in both weight (FCG-$W$) and inducing space (FCG-$U$) as well as Ensemble-$U$ better captures the increased predictive variance, although the mean function is more similar to that of FFG-$W$.

## 4.2 CLASSIFICATION AND IN-DISTRIBUTION CALIBRATION

As the core empirical evaluation, we train Resnet-18 models (He et al., 2016b) on CIFAR-10 and CIFAR-100 (Krizhevsky et al., 2009). To avoid underfitting issues with FFG-W, a useful trick is to set an upper limit $\sigma^2_{max}$ on the variance of $q(W)$ (Louizos & Welling, 2017). This trick is similarly applied to the $U$-space methods, where we cap $\lambda \leq \lambda_{max}$ for $q(W|U)$, and for FFG-$U$ we also set $\sigma^2_{max}$ for the variance of $q(U)$. In convolution layers, we treat the 4D weight tensor $W$ of shape $(c_{out}, c_{in}, h, w)$ as a $c_{out} \times c_{in}hw$ matrix. We use $U$ matrices of shape $128 \times 128$ for all layers (i.e. $M = M_{in} = M_{out} = 128$), except that for CIFAR-10 we set $M_{out} = 10$ for the last layer.

In Table 2 we report test accuracy and test expected calibration error (ECE) (Guo et al., 2017) as a first evaluation of the uncertainty estimates. Overall, Ensemble-$W$ achieves the highest accuracy, but is not as well-calibrated as variational methods. For the inducing weight approaches, Ensemble-$U$ outperforms FFG-$U$ on both datasets. It is overall the best performing approach on the more challenging CIFAR-100 dataset (close-to-Ensemble-$W$ accuracy and lowest ECE).

In Figure 4 we show prediction run-times on trained models, relative to those of an ensemble of deterministic networks, as well as relative parameter sizes to a single ResNet-18. The extra run-time costs for the inducing methods come from computing the extended Matheron's rule. However, as they can be calculated once and then cached when drawing multiple samples, the overhead reduces to a small factor when using larger number of samples $K$ and large batch-size $N$. More importantly, when compared to a deterministic ResNet-18 network, the inducing weight models reduce the parameter count by over 50% ($5,352,853$ vs. $11,173,962$, $47.9\%$) even for a large $M = 128$.

**Hyper-parameter choices** We visualise in Figure 5 the accuracy and ECE results for the inducing weight models with different hyper-parameters. It is clear from the right-most panels that performances in both metrics improve as the $U$ matrix size $M$ is increased, and the results for $M = 64$ and $M = 128$ are fairly similar. Also setting proper values for $\lambda_{max}, \sigma_{max}$ is key to the improved performances. The left-most panels show that with fixed $\sigma_{max}$ values (or with ensemble in $U$ space), the preferred conditional variance cap values $\lambda_{max}$ are fairly small (but still larger than 0 which corresponds to a point estimate for $W$ given $U$). For $\sigma_{max}$ which controls variance in $U$ space, we see from the top middle panel that the accuracy metric is fairly robust to $\sigma_{max}$ as long as $\lambda_{max}$ is not too large. But for ECE, a careful selection of $\sigma_{max}$ is required (bottom middle panel).

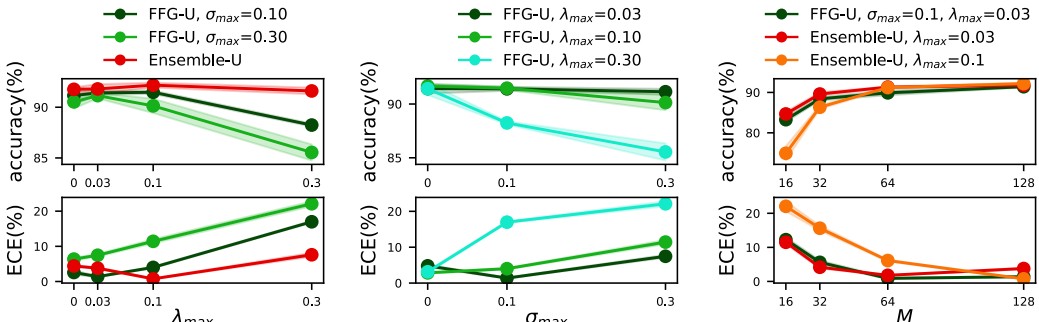

Figure 5: Averaged CIFAR-10 accuracy ($\uparrow$) and ECE ($\downarrow$) results for the inducing weight methods with different hyper-parameters. Models reported in the first two-columns uses $M = 128$ for $U$ dimensions. For $\lambda_{max} = 0$ (and $\sigma_{max} = 0$) we use point estimates for the corresponding variables.

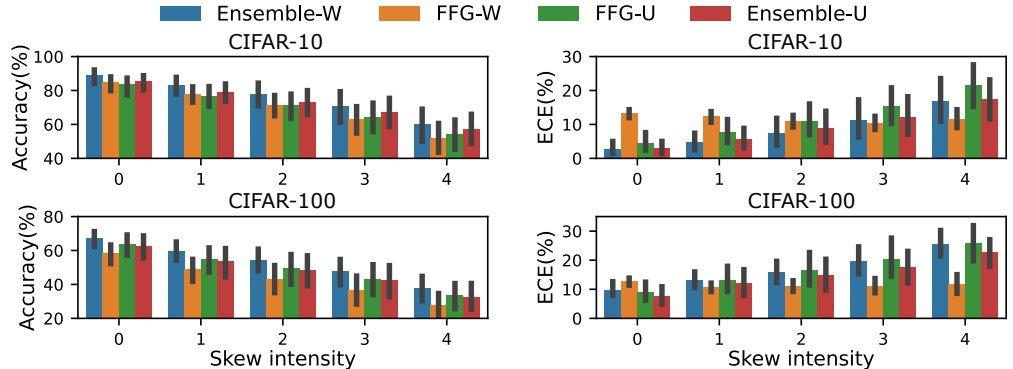

Figure 6: Accuracy ($\uparrow$) and ECE ($\downarrow$) on corrupted CIFAR. We show the mean and two standard errors for each metric on the 19 perturbations provided in (Hendrycks & Dietterich, 2019).

### 4.3 MODEL ROBUSTNESS AND OUT-OF-DISTRIBUTION DETECTION

To investigate the inducing weight's robustness to dataset shift, we compute predictions on corrupted CIFAR datasets (Hendrycks & Dietterich, 2019) after training on clean data. Figure 6 shows accuracy and ECE results. Ensemble-$W$ is the most accurate model across skew intensities, while FFG-$W$, though performing well on clean data, returns the worst accuracy under perturbation. The inducing weight methods perform competitively to Ensemble-$W$, although FFG-$U$ surprisingly maintains slightly higher accuracy on CIFAR-100 than Ensemble-$U$ despite being less accurate on the clean data. In terms of ECE, the inducing weight methods again perform competitively to Ensemble-$W$, with Ensemble-$U$ sometimes being the best among the three. Interestingly, while the accuracy of FFG-$W$ decays quickly as the data is perturbed more strongly, its ECE remains roughly constant.

We further present in Table 3 the utility of the maximum predicted probability for out-of-distribution (OOD) detection when presented with both the in-distribution data (CIFAR10 and CIFAR100 test sets) and an OOD dataset (CIFAR100/SVHN and CIFAR10/SVHN). The metrics are the area under the receiver operator characteristic (AUROC) and the area under the precision-recall curve (AUPR). Again Ensemble-$W$ performs the best in most settings, but more importantly, the inducing weight methods achieve very close results despite using the smallest number of parameters.

## 5 RELATED WORK

**Parameter-efficient uncertainty quantification methods** Recent research has proposed Gaussian posterior approximations for BNNs with efficient covariance structure (Ritter et al., 2018; Zhang et al., 2018b; Mishkin et al., 2018). The inducing weight approach differs from these in introducing structure via a hierarchical posterior with low-dimensional auxiliary variables. Another line of work reduces the memory overhead via efficient parameter sharing (Louizos & Welling, 2017; Wen et al.,

Table 3: OOD detection metrics for Resnet-18 trained on CIFAR10/100.

| In-dist. | CIFAR10 | | | | CIFAR100 | | | |
|---|---|---|---|---|---|---|---|---|
| OOD | CIFAR100 | | SVHN | | CIFAR10 | | SVHN | |
| Method / Metric | AUROC | AUPR | AUROC | AUPR | AUROC | AUPR | AUROC | AUPR |
| Deterministic-$W$ | $.87{\pm}.00$ | $.86{\pm}.00$ | $.92{\pm}.01$ | $.88{\pm}.02$ | $.73{\pm}.00$ | $.76{\pm}.00$ | $.80{\pm}.00$ | $.72{\pm}.01$ |
| Ensemble-$W$ | $.89$ | $.91$ | $.95$ | $.94$ | $.77$ | $.80$ | $.85$ | $.77$ |
| FFG-$W$ | $.87{\pm}.00$ | $.89{\pm}.00$ | $.89{\pm}.01$ | $.86{\pm}.01$ | $.75{\pm}.00$ | $.78{\pm}.00$ | $.79{\pm}.02$ | $.67{\pm}.04$ |
| FFG-$U$ | $.86{\pm}.00$ | $.88{\pm}.00$ | $.90{\pm}.00$ | $.87{\pm}.01$ | $.77{\pm}.00$ | $.79{\pm}.00$ | $.84{\pm}.01$ | $.76{\pm}.01$ |
| Ensemble-$U$ | $.86{\pm}.00$ | $.88{\pm}.00$ | $.89{\pm}.01$ | $.84{\pm}.02$ | $.77{\pm}.00$ | $.80{\pm}.00$ | $.83{\pm}.00$ | $.74{\pm}.01$ |

2020; Swiatkowski et al., 2020; Dusenberry et al., 2020). They all maintain a "mean parameter" for the weights, making the memory footprint at least that of storing a deterministic neural network. Instead, our approach shares parameters via the augmented prior with efficient low-rank structure, reducing the memory use compared to a deterministic network. In a similar spirit to our approach, Izmailov et al. (2019) perform inference in a $d$-dimensional sub-space obtained from PCA on weights collected from an SGD trajectory. However, this approach does not leverage the layer-structure of neural networks and requires $d\times$ memory of a single network.

**Sparse GP and function-space inference**  As BNNs and GPs are closely related (Neal, 1995; Matthews et al., 2018; Lee et al., 2018), recent efforts have introduced GP-inspired techniques to BNNs (Ma et al., 2019; Sun et al., 2019; Khan et al., 2019; Ober & Aitchison, 2020). Compared to weight-space inference, function-space inference is appealing since its uncertainty is more directly relevant for many predictive uncertainty estimation tasks. While the inducing weight approach performs computations in weight-space, Section 3.3 establishes the connection to function-space posteriors. Our approach is related to sparse deep GP methods with $U_c$ having similar interpretations as inducing outputs in e.g. Salimbeni & Deisenroth (2017). The major difference is that $U$ lies in a low-dimensional space, projected from the pre-activation output space of a network layer.

**Priors on neural network weights**  Hierarchical priors for weights has also been explored (Louizos et al., 2017; Krueger et al., 2017; Atanov et al., 2019; Ghosh et al., 2019; Karaletsos & Bui, 2020). However, we emphasise that $\tilde{p}(W, U)$ is a pseudo prior that is constructed to assist posterior inference rather than to improve model design. Indeed, parameters associated with the inducing weights are optimisable for improving posterior approximations. Our approach can be adapted to other priors, e.g. for a Horseshoe prior $p(\theta, \nu) = p(\theta|\nu)p(\nu) = \mathcal{N}(\theta; 0, \nu^2)C^+(\nu; 0, 1)$, the pseudo prior can be defined as $\tilde{p}(\theta, \nu, a) = \tilde{p}(\theta|\nu, a)\tilde{p}(a)p(\nu)$ such that $\int \tilde{p}(\theta|\nu, a)\tilde{p}(a)da = p(\theta|\nu)$. In general, pseudo priors have found broader success in Bayesian computation (Carlin & Chib, 1995).

## 6 CONCLUSION

We have proposed a parameter-efficient uncertainty quantification framework for neural networks. It augments each of the network layer weights with a small matrix of inducing weight, and by extending Matheron's rule to matrix-normal related distributions, maintains a relatively small run-time overhead as compared with ensemble methods. Critically, experiments on prediction and uncertainty estimation tasks demonstrate the competence of the inducing weight methods to the state-of-the-art, while reducing the parameter count to less than half of a deterministic ResNet-18.

Several directions are to be explored in the future. First, modelling correlations across layers might further improve the inference quality. We outline an initial approach leveraging inducing variables in Appendix E. Second, based on the function-space interpretation of inducing weights, better initialisation techniques can be inspired from the sparse GP and dimension reduction literature. Lastly, the small run-time overhead of our approach can be mitigated by a better design of the inducing weight structure as well as vectorisation techniques amenable to parallelised computation.

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
