# OpenReview forum: "Sparse Uncertainty Representation in Deep Learning with Inducing Weights"
_ICLR.cc/2021/Conference — Reject_

### Official Review · AnonReviewer3 · 2020-10-26
**Unclear gain in knowledge**

**Rating:** 5
**Confidence:** 2

**Review:**

The paper proposes to define the weights approximate posterior of a NN with inducing variables.

The main benefit of the approach lies in the compactness of the description (Fig4, right). However, if memory size is the main issue, entropy should be considered instead of just a count of parameters. Moreover, if complexity and memory footprint is the main concern, MC dropout sounds like a reasonable alternative to Bayesian NN. Why do the authors do not compare their proposal to a conventional MC dropout approach?

Overall, the contribution of the paper appears to be valid but relatively limited in scope compared to what exists in the literature (inducing variables are known, Bayesian NN are known, ensemble methods are known). What is the gain in knowledge brought by the paper? I might have missed a piece of the contribution, but the way the paper is written does not help. It is not self-contained (many previous works have to be read to follow the story), and remains relatively opaque to the non-expert. It lacks a clear and eye-bird picture of the approach (including both training and inference steps), to position and compare it to existing works.

The overhead in optimizing (1) or (2), compared to training a deterministic NN, is not discussed.

Variables d_in^l is used in Section 2, but only defined in Section 3.

---

> ### Author Response · Authors · 2020-11-16
> **Response to Reviewer3**
>
> Thank you very much for your review. While the other reviewers have found the technical exposition novel and clear enough, it is valuable feedback that to the non-expert reader the paper is difficult to follow. Our aim is of course to make our work accessible to a more general machine learning audience than just the Bayesian deep learning research community. We appreciate that the write-up is dense in its current state and we already had to move some technical material to the appendix that we would have liked to include in the main text. Given that we now have an additional page available, we will include some more high-level discussion in our next revision. However, we hope you understand that it is not feasible to write a self-contained research paper within nine pages and we have to refer the reader to other works in some places.
>
> We do not consider MC dropout as a baseline because it does not reduce the parameter count compared to a deterministic network. One would have to either store the full set of trained weights and perform dropout at test time, which would require storing the same number of parameters as a deterministic network, or one would have store an ensemble of smaller networks where the weights that have been zeroed-out by dropout have been removed, which, depending on the dropout rate, would quickly exceed the storage requirements of a deterministic network. Further, dropout is not used in all architectures, for example Resnets do not use dropout, limiting its applicability compared to deep ensembles.
>
> While our work builds on existing methods, we believe -- in which the other reviewers (as experts in Bayesian deep learning) also agreed -- that our approach is highly novel, and our contribution is significant. In addition to our empirical evaluations, we make the following original technical contributions:
>
> * An efficient approximate inference method for Bayesian neural networks that uses **fewer** parameters than a deterministic network. To our knowledge, this is the first such method that has been applied successfully to modern NN architectures.
> * An extension of Matheron’s rule to efficiently sample from conditional Gaussian distributions that have a joint covariance matrix with Khatri-Rao product structure. This may be of use beyond variational inference and Bayesian neural networks.
> * An interpretation of the proposed inducing weight approach as related to function-space inference and (deep) Gaussian processes. To our knowledge, this is the first approach that explicitly introduces the inducing variable method (which has been hugely successful in Gaussian process literature) to deep neural networks, and our inducing weight method nicely connects weight-space and function-space methods for uncertainty estimation in deep learning.
>
>
> We hope this has clarified the contribution of our work and that you will consider adjusting your score.

---

> > ### Comment · AnonReviewer3 · 2020-11-24
> > **Why is it important to the field?**
> >
> > Thank you for your reply.
> > My rating has now been increased to reflect the value of the work carried out to improve this manuscript during the rebuttal. However, I keep it slightly below threshold because the text remains unclear regarding why this work is important to the field.
> > To my view, telling that this is the first time that inducing variables have been successfully used in DL is not enough.
> > Which kind of DL user might be interested in the solution proposed by the paper? As suggested in my initial comment, if the benefit is 'just' to give access to a lightweight formulation of a neural network providing uncertainty estimation, the authors should position their work with respect to alternative solutions, including pruned/quantized  networks (combined with proper uncertainty estimation approaches).

---

> > > ### Author Response · Authors · 2020-11-24
> > > **Parameter efficiency is a core research direction towards practical applicability of BNNs**
> > >
> > > Thank you for engaging in the discussion. As previously stated, we highly appreciate your feedback for making our work more accessible to a broader machine learning audience.
> > >
> > > While we are indeed proposing an ultimately practically-minded approach for overcoming memory cost as one of the major obstacles to deploying BNNs in production systems, our paper still primarily aims to **advance core research in Bayesian deep learning**. To that end, we believe that demonstrating how to leverage inducing variables -- a widely popular technique for scaling GPs to large datasets -- for inference in deep networks is in itself a contribution of significant interest to many readers. Indeed, the other reviewers have expressed their excitement in their reviews, e.g. Reviewer4 writes: “...given that the inducing points approach has achieved big successes for scalable GPs, I think the proposed method could be impactful for uncertainty modelling in deep learning models”.
> > >
> > > Our approach achieves a parameter count below the arguably most important threshold for BNNs -- that of a deterministic network -- while maintaining competitive predictive performance against **strong** baselines on modern deep architectures. To our knowledge, our work is the first successful demonstration of applying inducing variable methods towards parameter-efficient uncertainty estimation for modern architectures, and we believe our approach is highly likely to inspire future research.
> > >
> > > Pruning methods, in contrast, are mostly tied to variational mean-field and of independent interest for deterministic networks (hence ensembles), so a direct comparison would mostly be relevant to a particular combination of real-world dataset and network architecture rather than a more general research paper such as ours. Also, as an important note, we have achieved the milestone of reducing the parameter count compared to a deterministic network with the simplest scheme imaginable for setting the number of inducing rows and columns: a constant value for every layer across the network. We see multiple directions for future research towards further reducing the parameter count, e.g. designing an adaptive scheme for choosing the inducing weight dimensions, and a post-hoc pruning of the inducing weights.
> > >
> > > More broadly, there is a pressing need for good uncertainty estimation to ensure robustness and safety in many practical settings (medical diagnosis, self-driving cars, ...). However, the increased parameter count of existing BNN/deep ensemble approaches is one of the core issues preventing most techniques from being used in practice, especially given the fact that modern deep learning models are still rapidly increasing in size. In this regard, we believe that our work is an important first step towards parametrically efficient uncertainty estimation in NNs, a direction that most DL users are interested in.
> > >
> > > We hope this further clarifies where we see the contribution of our work. Thank you again for expanding on your review.

---

### Official Review · AnonReviewer2 · 2020-10-26
**Interesting work, some key baselines are missing**

**Rating:** 6
**Confidence:** 4

**Review:**

### Summary
This work proposes a specific parametrisation for the Gaussian prior and approximate posterior distribution in variational Bayesian neural networks in terms of inducing weights. The general idea is an instance of the sparse variational inference scheme for GPs proposed by Titsias back in 2009; for a given model with a prior p(W) perform variational inference on an extended model with a hierarchical prior p(U) p(W | U), that has the same marginal p(W) = \int p(U)p(W | U)dU as the original model. The authors then consider “U” to be auxiliary weights that are jointly Gaussian with the actual weights “W” and then use the decomposition p(W|U)p(U), q(W|U)q(U) for the prior and approximate posterior (which can easily be computed via the conditional Gaussian rules). Furthermore, they “tie” (almost) all of the parameters between q(W|U) and p(W|U) (similarly to Titsias, 2009). The main benefit from these two things is that since the mean and covariance of the Gaussian distribution over W conditioned on U can be efficiently represented as functions of U, whenever dim(U) << dim(W) we get reductions in memory for storing the distributions over the parameters in the network. The authors furthermore, discuss how to efficiently parametrize the joint distribution over W, U, discuss different choices for q(U) (that can lead to either traditional VI or something like deep ensembles). In addition, they also discuss how more efficient sampling from q(W|U) can be realised via an extension of the Matheron’s rule to the case of matrix random variables. Finally, they evaluate their method against traditional mean field variational Bayesian neural networks and deep ensembles on several tasks that include regression, classification, calibration and OOD performance.

### Pros
- The method provides a novel way to induce parameter efficiency in variational bayesian neural networks
- It connects to the sparse GP literature and the trick seems to be general enough, in that it can be applied to any distribution that admits some parametrisation in terms of a Gaussian random variable.
- The extension of the Matheron’s rule can be of independent interest
- Extensive set of experimental tasks, with a nice ablation study for lambda_max and sigma_max

### Cons
- Comparison against alternative approaches that induce parameter efficiency are missing
- Results are mixed and sometimes not very convincing


### Recommendation
While I find the main idea very interesting and the presentation of it relatively clear, I unfortunately cannot recommend acceptance of this work as is. The main motivation behind improving parameter efficiency can also be performed with other, perhaps much simpler, ways and comparisons against such approaches is missing. Furthermore, the results, at least in their current state, are a bit mixed and thus not very convincing.

### Detailed feedback
Overall, this work is interesting and relatively easy to follow. Using inducing variables in neural networks in not a new concept, as it has been used before in, e.g., [1, 2], but the motivation of using them as a means towards reducing the number of parameters of each distribution is new. The authors explain the main idea behind them in a clear manner. Furthermore, they clearly explain the need for using matrix normal distributions, explaining how the Kronecker product factorisation of the covariance improves the parametrization efficiency, along with how efficient sampling from q(W|U) can be performed. In addition, performing parameter efficient deep ensembles in U space is a nice bonus of this formulation. Section 3.3 however is dense and can be a bit hard to follow (whereas Appendix D is clearer). I would therefore suggest that the authors briefly describe the main idea in a couple of sentences and refer the readers to Appendix D instead.

My main point for feedback is for the experimental section. The authors argue at the beginning of section 4 that “the goal is to demonstrate competitive performance …. While remaining computationally efficient.” Is that claim for the training or evaluation phases? From what I see, during the training phase, the inducing weights framework is not faster or more efficient than the baselines. The main advantage of the inducing weight is instead memory reduction in terms of parameters which translates to “memory efficiency” and not “computational “efficiency”. If then memory efficiency is the main target, I believe that some reasonable baselines are missing. For the case of a variational Bayesian neural network a simple baseline that performs pruning post-hoc (e.g., the one presented at the FFG-W paper or the one from [3]) in order to reduce the parameter count; remove weights by either setting them to exact zero or equal to the prior and then use the variational posterior for those that survive. It would be interesting to see how such an approach would fare not just on accuracy, but also on the ECE and OOD. Similarly for deep ensembles, a baseline where you perform pruning (e.g., simple magnitude based) would also serve as a better baseline for the Ensemble-U.  Both of these baselines would provide a more complete picture and would be a better signal in understanding whether the inducing weights framework is a better choice overall.

As for other points

- At figure 3 you only show FCG-U and not the FFG-U which is used in all of the other experiments. How does the uncertainty look like with FFG-U?
- Sometimes, both FFG-U and Ensemble-U underperform compared to FFG-W and Ensemble-W and it would be good to know if it is due to less free parameters to optimise or due to the model definition itself. How does FFG-U perform when increasing dim(U) such that the parameter count is the same as FFG-W? Similarly for Ensemble-U vs Ensemble-W.
- The second point at the end of page 2 only makes sense in hindsight (i.e., after one reads section 3). Perhaps the authors could expand on them a bit better so that it is self-contained (e.g., explain what you mean with “q(theta|alpha) can be efficiently parametrised”).


[1] Structured and efficient variational deep learning with matrix Gaussian posteriors, C. Louizos & M. Welling, 2016
[2] Global inducing point variational posteriors for Bayesian neural networks and deep gaussian processes, S. Ober & L. Aitchison, 2020
[3] Practical Variational Inference for Neural Networks, A. Graves, 2011

---

> ### Author Response · Authors · 2020-11-16
> **Response to Reviewer2 (2/2)**
>
> Regarding your other points:
> * We will add the corresponding figure to the appendix. The uncertainty is similar to FFG-W.
> * Overall we would say that the performance differences are rather small. The main factor that determines the number of parameters is the dimensionality of the inducing space. We only ran our ablation study for the dimensionality of the inducing space up to half the parameters of the deterministic network (or a quarter of the number of mean-field parameters), but given that both accuracy and ECE start to saturate (Figure 5; right column) at the value of M=128 that we use in our main figures and tables, we would expect that matching the number of parameters would at most give a marginal improvement for the performance of the inducing weight methods.
> * Thanks for pointing this out. We had to move more material than we would have liked to the appendix to be within the page limit, but will upload a revised version that makes use of the additional page that is now available. Similarly we will revise section 3.3 to make it clearer.

---

> > ### Comment · AnonReviewer2 · 2020-11-23
> > **Response to rebuttal**
> >
> > I would like to thank you for addressing my comments and since the clarity has improved I will raise my score.
> >
> > Nevertheless, I still cannot fully recommend acceptance, as I would still argue in favour of the baselines I mentioned. Both this framework and, e.g., FFG-W are doing inference under the same normal prior (i.e., they have the same model) and parameter reductions come essentially from the parametrisation / sparsity of the variational distribution. Furthermore, in my opinion, the primary motivation for reducing parameter count would be to facilitate for practical deployment of the BNN in cases where we also require uncertainty (e.g., a self driving car). As a result, since mean-field variational inference is known to produce pruning [1], it would be a valid baseline to include. In general, if  parameter count during training is an issue there are also alternative ways (which in a sense are simper) that can be employed, namely data parallelising across multiple gpu’s and / or using gradient checkpointing.
> >
> > [1] Overpruning in Variational Bayesian Neural Networks, Brian Trippe, Richard Turner, 2017

---

> > > ### Author Response · Authors · 2020-11-24
> > > **Re: response**
> > >
> > > Thank you for further elaborating on your review. Scientifically, we still believe that comparing to the performance of mean-field pruning would not add much insight into **our** method to the paper and that the most important milestone to pass is having fewer parameters than a deterministic network. As the discussion period is almost over and this appears to be a core concern to you, we will investigate how different levels of pruning affect the predictive performance of FFG-W for the camera-ready revision. We hope this puts you more at ease with recommending acceptance.
> > >
> > > We are primarily interested in reducing the parameter count at test time, so parallelising across GPUs and gradient checkpointing are not applicable, although they are of course compatible with our method for training.
> > >
> > > Thank you again for your review and engaging in the discussion.

---

> ### Author Response · Authors · 2020-11-16
> **Response to Reviewer2 (1/2)**
>
> Thank you very much for your extensive and thorough view, in particular your feedback regarding the experimental section and suggesting specific baselines to compare against. We had indeed considered comparing to methods from the literature on pruning neural network weights but ultimately decided against this.
>
> Our reasoning is that such a comparison would mainly be needed if we were presenting our method as essentially ready-to-deploy for e.g. machine learning on mobile devices, which would indeed require pushing the memory requirement to be **as small as possible**. We believe that none of the existing BNN/deep ensemble approaches aim at solving this challenge, in this regard the focus of our work is no different to existing paradigms. For example, Swiatkowski et al. (2020) recently proposed an efficient parameterisation of the mean-field posterior that reduces the number of parameters to be slightly larger than that of a deterministic network (down from twice as many parameters) without a comparison to pruning methods, which would in all likelihood yield a greater reduction since a pruning rate of 50% would be sufficient to save more parameters that their method.
>
> However, we do notice that BNN/deep ensembles need to reduce their memory overhead to a **reasonable level** in order to be scaled up to giant neural networks such as very deep Resnets and Transformers. Therefore, we see our contribution mainly on the conceptual side, showing that we can efficiently perform inference in a lower-dimensional space while having a smaller parameter count than the dimensionality of the original space. We strongly believe that our experiments support this claim and our initial results on Resnet50 further confirm the efficacy of our approach (accuracy of 94.72% and ECE of 1.61% for FFG-U). As a side note, subspace methods have been shown to be effective in the literature before, e.g. (Izmailov et al., 2019), however all of these approaches pay the cost of **increasing** the number of parameters.
>
> The variational mean-field pruning that you suggest is a reasonable approach to further reduce parameter count. However, in contrast to our inducing weight method, it **changes the model post-hoc**, whereas we perform inference in the original model, so it would not be an apples-to-apples comparison. Instead, pruning our inducing-weight based network post-hoc would be an interesting direction for a fair comparison in future work.
>
> On another important note, we have not in any way pushed for the minimum attainable number of parameters with our inducing weight method. In fact, we have used the simplest scheme imaginable for setting the number of inducing rows and columns: a constant value for every layer across the network, as this allows for an illustrating ablation study and greatly simplifies the experimental setup. The alternative would be to devise a scheme for adapting shape($U_l$)  to the specific network architecture and the learning task. In this regard we see the potential to further reduce the memory requirements of our approach. However, we do not believe that this would add particularly important insights to the paper.
>
> Finally, there is no established experimental benchmark for the parameter efficiency of BNNs, and we do not think that it makes a significant difference to create such a protocol at this point given that existing methods are not routinely used for the purpose of on-device ML. Even if such a benchmark existed, the ICLR reviewer guidelines explicitly state papers do not need to achieve state-of-the-art results for acceptance, but need to bring “new, relevant, impactful knowledge” to the community. Based on the feedback both from you and the other reviewers, we are confident that we have achieved this.
>
> We hope you agree and will consider increasing your score, otherwise we are more than happy to engage in further discussion.
>
>
> References:
>
> Swiatkowski et al. The k-tied Normal Distribution: A Compact Parameterization of Gaussian Mean Field Posteriors in Bayesian Neural Networks. In ICML 2020.
>
> Izmailov et al. Subspace inference for Bayesian deep learning. In UAI 2019.

---

### Official Review · AnonReviewer4 · 2020-10-27
**A compact variational BNN posterior for memory efficiency**

**Rating:** 6
**Confidence:** 4

**Review:**

Bayesian deep learning attempts to incorporate uncertainty estimation in the modern neural network models. However, common approaches such as Bayesian neural networks and Deep Ensembles incur large memory overhead because of the increased parameter sizes. Motivated by the inducing points approach in the sparse variational Gaussian Processes area, this paper proposes the Gaussian variational posterior relying on the compact *inducing weights* $U$ and the conditional distribution $p(W|U)$. Because the variational parameter shifts from $W$ to the smaller $U$, the resulting model potentially uses even fewer parameters than a deterministic counterpart. In general, I think the paper proposes an interesting approach that could help addressing the storage issues of current Bayesian deep learning models.

**Novelty**
The proposed approach is both novel and elegant. Given the high dimensionality of the parameters, many BNN approaches consider structured covariance for the variational posterior, which involves the balancing between computational costs and approximating capacity. And these approaches all maintain a "mean parameter", thus the memory costs are at least as large as the original network. In comparison, this paper turns to an augmented space using the joint Gaussian distribution, and instead model the variational posterior for the compact *inducing weights*. In consequence, the resulting model could potentially have even fewer parameters than a deterministic counterpart, while being able to conduct uncertainty quantification. Given that the inducing points approach has achieved big successes for scalable GPs, I think the proposed method could be impactful for uncertainty modelling in deep learning models.

**Experiments**
This paper conducts experiments covering both image classification and out-of-distribution detection. Empirically, their model has increased calibration compared to the deterministic network while using $\leq 47.9\%$ of parameters. However, as shown in Figure 4, the proposed approach incurs large computational overheads especially when using small number of samples.

**Questions**
1. What are the inducing weights for the conv layer? Are they matrices of shape [128, 128, kern_size, kern_size]?
2. The Ensemble-U approach uses a dirac measure $\sum_k \delta_{U^k_l}$ in each layer and drops the KL penalty. Are dirac measures of different layers independent, or like deep ensemble, the particles are tied into $K$ disjoint groups across layers? If it was the former, I would reckon that the particles of the same layer would converge to the same place; if it was the latter, it should be made clearer in the paper.
3. How does the sample size $K$ influence the performance? In practice, I think only a small number of $K$ should be used.
4. A hyperparameter $\sigma_{max}^2$ is set for the variance of $q(W)$. And the similar hyperparameter $\lambda_{\max}$ is set for the variational posterior. Especially for $\lambda_{max}$, increasing it from $0.1$ to $0.3$ observes large performance drop. Is this an issue due to the expressiveness of the proposed posterior ?

**Clarity**
In spite of my questions regarding the model details, the paper well presents its main methods. Besides, I think Sec3.3 is worthy of more polishing, since it looks confusing when you start by studying $U_c$ instead of $U_r$ or $U$. And a few typos to be corrected, 1. Abstract: whichenable 2. Last paragraph of introduction: our approach achieve 3) Related works: function-space inference is appealing to ...

---

> ### Author Response · Authors · 2020-11-16
> **Response to Reviewer4**
>
> Thank you very much for your review. We are pleased to see that you appreciate the novelty and the technical contributions of our paper. We will address your questions below:
>
> 1. We treat convolutional weights as 2d matrices by ‘flattening’ the 4d tensors from shape (out_channels, in_channnels, height, width) to (out_channels, in_channels * height * width). This corresponds to the interpretation of convolutions as linear layers that are repeated across the image patches with tied weights, see e.g. (Grosse & Martens, 2016) for an equivalent use for second-order optimisation. We will clarify this in our next revision of the paper.
>
> 2. We indeed use the latter interpretation, i.e. tie the groups across layers. Again, we will update the paper to better reflect this.
>
> 3. Similar to previous work, e.g. (Ovadia et al., 2019), we found in preliminary experiments a small sample size of around 5 to be sufficient for capturing most of the performance gains over using a single sample. We used 20 samples for all variational methods, as we believe that this fairly represents the advantage over deterministic ensembles of being able to draw an arbitrary number of samples after training.
>
> 4. This is an interesting question and we don’t have a conclusive answer at this point. It has recently been argued in the BNN literature that a more concentrated ‘cold posterior’ (Wenzel et al., 2020) can give better predictive performance and we would hypothesise that our observations regarding the hyperparameters are related to this phenomenon. Further it has been observed that more expressive variational posteriors can lead to worse performance (Trippe & Turner, 2017), however since our approximate posterior has a low-rank mean, but a non-diagonal covariance, it is difficult to strictly classify it as more or less expressive than e.g. a mean-field posterior. Overall, we think that further research around the true posterior of deep neural networks is needed and that our low rank approach in the original model makes for an interesting contribution in that direction.
>
> We are glad to see from your comments that the main message of our paper is clearly conveyed. Still we will revise our paper, especially for section 3.3, to further improve clarity.
>
>
> References:
>
> Grosse & Martens. A Kronecker-factored approximate Fisher matrix for convolution layers. In ICML 2016.
>
> Ovadia et al. Can You Trust Your Model’s Uncertainty? Evaluating Predictive Uncertainty Under Dataset Shift. In NeurIPS 2019.
>
> Wenzel et al. How Good is the Bayes Posterior in Deep Neural Networks Really? In ICML 2020.
>
> Trippe & Turner. Overpruning in Variational Bayesian Neural Networks. In NeurIPS 2017 Workshop on Advances in Approximate Bayesian Inference.

---

> > ### Comment · AnonReviewer4 · 2020-11-20
> > **Response**
> >
> > Thank you so much for the responses. I think your responses resolve most of my concerns. However, as I mentioned in the **Experiments** section, I think the runtime-overhead is a major issue of the propose approach. Given that you used K=5 over your experiments, the proposed approach seems to be ~10x slower as shown by Figure 4. Could you please comment more on this ?

---

> > > ### Author Response · Authors · 2020-11-20
> > > **Runtime overhead is mitigated with large batchsize**
> > >
> > > Thank you for your response, glad to see that many of your previous concerns are resolved.
> > >
> > > Regarding runtime overhead:
> > >
> > > The time complexity figures in Table 1 shows that compared with FFG-W, FFG-U has extra overhead (the $\mathcal{O}(2 M^3_{in} + 2 M^3_{out} + K(d_{out} M_{out} M_{in} + M_{in} d_{out} d_{in}))$ term) required by computing the Cholesky decomposition and matrix multiplications.
> > >
> > > However, this extra cost dominates the computation **only when batch-size $N$ and number of samples $K$ are small**. We expect that when $N > M_{in}$ and $K$ is reasonably large, the dominate cost would be the regular $\mathcal{O}(NK d_{in} d_{out})$ term that also appears in FFG-W time complexity. This is because:
> > > 1. The Cholesky decompositions are only needed once (the $\mathcal{O}(2 M^3_{in} + 2 M^3_{out})$ term).
> > > 2. When $N > M_{in}$, we have $\mathcal{O}(NK d_{in} d_{out}) > \mathcal{O}(K(d_{out} M_{out} M_{in} + M_{in} d_{out} d_{in}))$, i.e. constructing $K$ set of network parameters becomes cheaper.
> > > 3. If time budget is in concern, we can always cache both the Cholesky decomposions and the network parameter constructions. In this case inference time complexity reduces back to $\mathcal{O}(NK d_{in} d_{out})$, but we need to pay the price of caching the full network parameters.
> > >
> > > You can see in Figure 4 (left) that when $N=500$ and $K > 8$, the run time overhead of FFG-U as compared with FFG-W becomes smaller and smaller, converging towards a relative runtime ratio of 1. In our test time evaluations we used $N=500$, $K=20$, and $M_{in} \leq 128$.
> > >
> > > We would like to emphasize again that our goal is to develop a **parameter efficient** approach for uncertainty estimation in deep neural networks. Still we agree that it would be useful to address the run-time overhead for this method. As we stated in the conclusion part of the paper, we believe some vectorisation technique, e.g. see Wen et al. (2020), could be develop to improve the speed.

---

> > > > ### Author Response · Authors · 2020-11-20
> > > > **Bigger network architecture also reduces overhead**
> > > >
> > > > To further elaborate on the point that the computational overhead decreases with batch size: this is also the case for a bigger architecture. We have just run our runtime benchmarks for Resnet50 and Resnet101 -- which besides being deeper, also have some wider layers compared to Resnet18 -- and the overhead for using FFG-U instead of FFG-W decreases from around $11\times$ to $7\times$ at a mini-batch size of $100$, and from around $4\times$ to $2.5\times$ at a batch a size of $500$ (all with $K=4$ samples and $M=128$). The overhead for sampling the weights becomes less and less significant as the computation time of the forward pass increases.

---

> > > > > ### Comment · AnonReviewer4 · 2020-11-22
> > > > > **Response**
> > > > >
> > > > > Thanks for your detailed responses ! Overall, I think your paper is very novel and interesting to read !

---

### Official Review · AnonReviewer1 · 2020-10-28
**Official Blind Review #1**

**Rating:** 6
**Confidence:** 4

**Review:**



# summary

This paper proposed a method on uncertainty estimation in deep neural
networks. Compared with BNN and deep ensemble, the proposed approach in this
work has a storage advantage. Furthermore, this work provides a better
trade-off between accuracy and calibration.


# pros

1.  The approach in this work is quite interesting. The idea of augmenting
    weights with auxiliary low-dimensional latent variables in a deep neural
    network seems natural at first sight, but this approach is novel as far as
    my knowledge is concerned. Although VI with local latent variables is an
    old technique, this application in deep neural network is novel.
2.  The authors also proposed an efficient approach that can sample from the
    variational approximation conditioning on the latent variable. Since the
    original weight is large, such a sampling is necessary.
3.  This paper provides extensive empirical results and sufficient theoretical
    results. Experimental results show the proposed approach achieve a good
    balance between accuracy, calibration and memory requirements.


# cons

1.  My major concern is all experiments are conducted on ResNet18, and this is
    not a typical choice for practical problems. I think experiments on other
    larger net such as ResNet56 on CIFAR10 will make this paper more
    convincing.
2.  It is not clear to me how the authors choose the "mixture of delta
    measures", i.e. how to choose U<sup>k</sup>? Can the authors comment on this? In
    this case, q(U) is a categorical distribution. It seems better if  q(U)
    also depends on input data, however, the authors choose to use a fixed
    q(U). Can the authors comment on this choice?

Overall speaking, the idea and the presentation of this work are great in my
opinion.

---

> ### Author Response · Authors · 2020-11-16
> **Response to Reviewer1**
>
> Thank you very much for your encouraging review. We are particularly pleased that you appreciate the novelty of the idea and the clarity of the presentation.
>
> **Architecture size** In order to avoid possible confusion regarding the architecture, we would like to emphasize that the Resnet18 architecture we use is based on the torchvision implementation, i.e. the smallest ImageNet-sized architecture (>11M parameters), **not** the CIFAR-sized Resnet20 architecture from the original Resnet paper (<1M parameters). The Resnet56 architecture you suggest was also developed for CIFAR (<1M parameters) -- did you mean Resnet50 (>22M parameters) or were you under the impression that we were using the smaller CIFAR architectures?
> While we fully agree that Resnet50 would be more representative of an architecture that a practitioner would use, we do believe that Resnet18 captures the most relevant properties of a modern deep neural network. For example, Osawa et al. (2019) also based their Bayesian deep learning experiments on Resnet18.
>
> We are currently running some preliminary experiments for FFG-W, FFG-U and Ensemble-U on Resnet50 for CIFAR10. Those take slightly more than twice as long compared to the Resnet18 experiments due to the larger network architecture. Given the short discussion period, we wish to provide initial results in response to your question, and later on we will update the full set of experiments in the camera-ready version. Based on our first runs, we can confirm that the inducing weight method works on the larger architecture and accuracy increases as expected while maintaining a low ECE. The results we have available for Resnet50 at this point are as follows:
>
> |Method | Accuracy(%) | ECE(%) | #params | %params of det. net (including BatchNorm) | #seeds |
> |---|---|---|---|---|---|
> |Deterministic | 94.47 | 4.59 | 23,520,842 | 100 | 5 |
> |FFG-U (M=128)| 94.72 | 1.61 | 12,253,366 | 52.1 | 3 |
> |FFG-U (M=64) | 94.33 | 0.66 | 5,710,902 | 24.28 | 2 |
> |FFG-W | 93.24 | 0.66 | 46,988,564 | 199.8 | 3 |
> |Ensemble-U (M=128) | 95.28 | 1.86 | 14,907,574 | 63.4 | 3 |
> |Ensemble-W (K=5) | 95.58 | 1.32 | 117,604,210 | 500 | 1 (from det. seeds)
>
> **Ensemble-U** By “mixture of delta measures” we mean having $k$ sets of continuous “parameter” values in inducing space. You can think of this as equivalent to a classical ensemble of neural networks, except that the ensemble is in the lower-dimensional U space and then each set of inducing weights is (stochastically) projected into the original parameter space for a separate forward pass (the projection parameters are shared).
>
> Having q(U) depend on the input would be a change of the model from globally shared parameters to parameters local to each data point as in the classical mixture density networks (Bishop, 1994). We have opted for the global weight model mostly to avoid an additional factor of variation between the two inference methods we compare. Introducing an input dependence would certainly be an interesting extension of our work and is a direction that has received attention recently e.g. in (Kristiadi et al., 2019) and (Karaletsos & Bui, 2020).
>
> We hope our response has been helpful and are of course happy to further clarify any additional questions you may have.
>
>
> References:
>
> Osawa et al. Practical Deep Learning with Bayesian Principles. In NeurIPS 2019.
>
> Bishop. Mixture density networks. 1994.
>
> Kristiadi et al. Predictive Uncertainty Quantification with Compound Density Networks. arxiv preprint arXiv:1902.01080.
>
> Karaletsos & Bui. Hierarchical Gaussian Process Priors for Bayesian Neural Network Weights. NeurIPS 2020.

---

### Author Response · Authors · 2020-11-16
**Summary of the response**

We thank all reviewers for their valuable feedback. We are pleased that reviewers 1, 2 and 4 find our work novel and interesting and its presentation clear. They further highlight that our extension of Matheron’s rule for sampling conditional Gaussian variables can be of independent interest. We have addressed minor comments in the individual responses and will upload a revision of the paper in the next few days.

Reviewer 1 questioned whether Resnet18 was reflective of an architecture that would be used in a practical setting. While we believe this to be the case (see the response to reviewer 1), we have run initial experiments on the larger Resnet50 architecture and can confirm that our inducing weight method still works as expected on this deeper network. Accuracies, ECE and parameter counts on Resnet50 are as follows:

|Method | Accuracy(%) | ECE(%) | #params | %params of det. net (including BatchNorm) | #seeds |
|---|---|---|---|---|---|
|Deterministic | 94.47 | 4.59 | 23,520,842 | 100 | 5 |
|FFG-U (M=128)| 94.72 | 1.61 | 12,253,366 | 52.1 | 3 |
|FFG-U (M=64) | 94.33 | 0.66 | 5,710,902 | 24.28 | 2 |
|FFG-W | 93.24 | 0.66 | 46,988,564 | 199.8 | 3 |
|Ensemble-U (M=128) | 95.28 | 1.86 | 14,907,574 | 63.4 | 3 |
|Ensemble-W (K=5) | 95.58 | 1.32 | 117,604,210 | 500 | 1 (from det. seeds)

We will update the full set of experiments for the camera-ready version and do not anticipate any significant changes of the results or conclusions.

Reviewer 2 expressed concerns about a lack of comparison to baseline approaches for (post-hoc) network pruning, but we don’t think this would give an apples-to-apples comparison (see response to reviewer 2). Nevertheless, we believe our approach has the potential to be pushed further for better memory savings. Indeed, we could have spent space in the manuscript discussing heuristics on choosing shape($U$) by adapting to the network architecture and the learning task. We decided not to push in this direction for the following reasons:

1. The goal of our work is to perform **uncertainty estimation** in deep neural networks with **acceptable** memory consumptions. This is important especially for giant neural networks since maintaining even a single copy of them would require massive storage already. Our approach provides the first solution towards reducing BNN/deep ensemble memory to **below** the memory of their deterministic counterpart, and we believe our current experiments clearly and convincingly demonstrate the efficacy of our approach.
2. We don’t see that pushing the parameter count to the extremely small side (e.g. < 1%) for a specific architecture on CIFAR10/100 would have added much insight to the paper. Instead, we believe that it is more valuable to keep the hyperparameter search space for shape($U$) simple, since we wish to design an experimental protocol that is easy to follow for the reader and that lends itself to ablation studies (Fig. 5). The positive responses from the reviewers regarding clarity has reinforced this belief.
3. To our knowledge, research on compressing neural networks has so far been focusing on maintaining **a minimum test accuracy** while reducing memory usage. Variational inference based techniques for network compression also focus on accuracy maintenance only, see e.g. (Louizos et al., 2017) and (Havasi et al., 2019). Investigating the maintenance of a minimum level of uncertainty calibration could be interesting future work, but again, we believe that adding this study to our paper is out of scope and would convolute the main message that we want to convey.


Again, we want to thank all of the reviewers and hope that we have now fully convinced all of them that our paper makes for a valuable contribution to the conference.


References:

Louizos et al. Bayesian compression for deep learning. NeurIPS 2017

Havasi et al. Minimal random code learning: Getting bits back from compressed model parameters. ICLR 2019

---

### Author Response · Authors · 2020-11-20
**New revision**

Dear reviewers, we have just uploaded a new revision of our paper. To highlight the changes, we have marked any new text in red font. Please find a comprehensive list of the most important updates below, indicating in brackets whose questions or suggestion we’re aiming to address:
* Explained overhead of training a BNN based on eq. 1 (R3)
* Defined variables d_in^l and d_out^l in Section 2 (R3)
* Clarified property 2 for the inducing auxiliary variables (R2)
* Expanded the introductory example of a multivariate Gaussian in Section 3.1
* Clarified what we mean by a “mixture of delta measures” for Ensemble-U in Section 3.1 (R1, R4)
* Significantly expanded Section 3.3 to make it less dense (R2)
* Added figures for FFG-U (and FCG-W) to the toy regression example in Section 4.1 (R2)
* Explained how we treat convolutional layers in Section 4.2 (R4)

We note that we have not included any preliminary results on Resnet50 in our current revision. We would like to finish our systematic run of all methods on both CIFAR10 and CIFAR100 with a sufficiently large number of random seeds before doing so, for which the discussion period has not been sufficiently long. We have updated the tables in our summary of the response and the reply to Reviewer1 to include additional seeds for FFG-U and FFG-W on CIFAR10 and we anticipate adding initial results for Ensemble-U when they are available. We believe that our preliminary results show that the inducing weight approach also works well on deeper networks such as Resnet50.

Thank you all for your valuable feedback. We will be happy to take any further suggestions into account.

---

### Author Response · Authors · 2020-11-23
**Updated results on Resnet50**

Dear reviewers, we have just updated our Resnet50 results for CIFAR10 in the "Summary of the response" below and the response to Reviewer1. We have added accuracy, ECE and parameter counts for deterministic networks, deterministic ensembles (Ensemble-W), our Ensemble-U method and FFG-U with a lower U dimension. In line with our results from the original submission for Resnet18, our inducing method achieves competitive predictive performance at a much reduced parameter count.

We will provide the full suite of Resnet50 results (additional random seeds; CIFAR100; uncertainty evaluation) in the camera-ready paper.

---

### Decision · Program_Chairs · 2021-01-07
**Final Decision**

**Decision:**

Reject

**Comment:**

The scores for this paper have been borderline, however the decision has been greatly facilitated by the participation of the authors and reviewers to the discussion and, more importantly, by active private discussion among reviewers and AC. Specifically, from the private discussion it seems that the reviewers find interesting ideas in this paper, but are overall are not entirely convinced about its significance, at least in the way the paper is currently positioned and motivated.

More specifically, the reviewers found the main idea of using inducing weights interesting, both technically (e.g. associated sampling scheme) but also in terms of application (sparsity). The results are insightful from a theoretical perspective. That is, the inducing weights and their treatment does seem to result in interesting and potentially useful statistical properties for the model. On the other hand, it is important to note that the high-level idea of variational inducing weights, with usage of matrix normals in this setting, as well as connection to deep GPs has been studied before, as pointed out by R2 (refs [1,2]). Furthermore, even after discussions the motivation is still not entirely convincing, especially in conjunction with the experiments. Although various interesting ideas exist in the paper, both R2 and R3 in particular remain unconvinced about what is the main benefit (e.g.  pruning or runtime efficiency) stemming out of the proposed idea. Another reviewer agreed with this point in the private discussions.  Apart from overall clearer positioning of the paper, the claimed benefit would need to be supported by experiments tailored to illustrate this main point. The authors argued against some of the suggested comparisons (e.g. past pruning methods), and further discuss that there is no established experimental benchmark for the parameter efficiency of BNNs. I indeed sympathize with both of these arguments; however, I believe that if the reviewers' suggestions for extra experiments are rejected, it remains the responsibility of the authors to find a slightly different way of motivating their work and demonstrating its efficiency in some convincing, meaningful and more well-defined setting with the appropriate benchmarks.